# Ocean-Bottom Laser Seismograph

**DOI:** 10.3390/s22072527

**Published:** 2022-03-25

**Authors:** Grigory Dolgikh, Stanislav Dolgikh, Aleksandr Plotnikov

**Affiliations:** V.I. Il’ichev Pacific Oceanological Institute FEB RAS, 690041 Vladivostok, Russia; sdolgikh@poi.dvo.ru (S.D.); lotos@poi.dvo.ru (A.P.)

**Keywords:** bottom laser seismograph, Michelson interferometer, wind wave, seiches, vertical bottom displacements, laser strainmeter

## Abstract

This paper describes an ocean-bottom laser seismograph, based on the modified laser meter of hydrosphere pressure variations, and designed to record vertical bottom displacements at the place of its location. Its measuring accuracy is about 1 nm, limited by the stability of the laser emission, which can be improved by using more advanced lasers. The purpose of this instrument is to measure the displacements of the seabed’s upper layer in the low-frequency sonic and infrasonic ranges. Theoretically, it can operate in the frequency range from 0 (conditionally) to 1000 Hz; the upper limit is determined by the operating speed of the digital registration system. We demonstrated the capabilities of the ocean-bottom laser seismograph while registering vertical bottom displacements caused by sea wind waves and lower frequency processes—seiches, i.e., eigenoscillations of the bay in which the instrument was installed. Comparison of experimental data of the bottom laser seismograph with the data of the laser hydrosphere pressure variations meter and the velocimeter—installed in close proximity—shows good efficiency of the instrument.

## 1. Introduction

Compared to dry land, the sea bottom has much larger area, but due to specific conditions (e.g., water, high pressure) it is practically inaccessible for placing modern instruments for registering wide-frequency-range signals. In view of these specific conditions, various systems have been developed for installation on the bottom.

These systems include, among others, ocean-bottom seismographs, developed by various firms and used for various purposes. A typical example is the ocean-bottom seismograph (OBS) [1], which can be used to study technogenic and natural earthquakes, the structure of the sea bottom and the Earth’s mantle, kinetic characteristics of crust plate shift zones and sea valleys, the seismic tomography of natural earthquakes, and earthquake forecasting. It has the following characteristics, which are almost classical for similar ocean-bottom seismographs: (1) operating frequency ranges of 4.5–300, 1–200, 0.2–150, 0.1–150, 0.03–150, 0.016–100, and 0.0083–50 Hz; (2) operational depth of up to 6000, 9000, and 12,000 m; (3) operational duration from 2 to 15 months; maximum time of operation from 6 to 24 months; and (4) three general types—broadband, narrowband, and portable. There is a combined ocean-bottom seismograph (COBS) [2] specifically designed to study microvibrations caused by earthquakes and other lithospheric structural anomalies. Its technical characteristics are as follows: (1) operating frequency range of 1–300 Hz; (2) operational depth of up to 1500 m; (3) operational duration of up to 30 days; and (4) can be converted into a floating ocean-bottom seismometer with a buoyant block, seismograph anchors, and acoustic release transponder. The author of [3] describes the seismographs of this type, which have a nonlinear amplitude–frequency characteristic in the frequency range from 0.002 to 100 Hz. In [4], other designs of ocean-bottom seismographs (self-floating and buoy-mounted) are presented, operating in different frequency ranges (5–125, 3–400, and 3–240 Hz) at depths of up to 6000–6700 m. Various organizations worldwide have developed and built various types of seismographs, albeit with practically the same technical characteristics—especially regarding operating frequency range—which have been successfully used to study signals on the ocean bottom in a limited frequency range; for example [5,6,7,8,9]. Data are collected from seismographs in two ways: data accumulation with subsequent interpretation after lifting, and data transmission via wired [10] and wireless [11] underwater networks.

The authors of [12] discussed the development of a broadband ocean-bottom seismograph (BBOBS) and its modernizations, i.e., the creation of a new-generation sensor (BBOBS-NX). The authors state that as the next step in seabed geophysical observation, the BBOBS and the BBOBS-NX could represent a breakthrough in establishing a geodesic observation network on the seabed. In recent years, vertical displacement observation has been widely performed using absolute pressure gauges, but other geodesic observations are rarely carried out. The broadband sensor in the BBOBS data has a mass position output signal, which can be used to measure tilt changes. As the horizontal component of the BBOBS-NX noise level is low in the far range, the authors state that it can be used to measure tilt. Such devices—i.e., tiltmeters, developed by various organizations—have been successfully used for the measurement of seabed tilts; for example [13].

Furthermore, we are not interested in the capabilities of instruments for seabed tilt measurement. We are primarily interested in the capabilities of instruments for measuring vertical displacements of the oceanic crust, caused by various processes of infrasonic and sonic ranges. If we consider the ocean-bottom seismographs described above, we can note their common drawback: all of them have a limited operating frequency range, especially in the infrasonic range. To overcome this drawback, we set the task of developing an ocean-bottom laser seismograph, based on modern laser-interference methods, which can significantly improve the characteristics of receiving systems. We named the developed system an ocean-bottom laser seismograph, which is not entirely correct. All seismographs, including the land-based ones, are inertia-type instruments, and the created ocean-bottom laser seismograph, based on the Michelson interferometer, is also an inertia-free instrument—more precisely, a power-type instrument.

Thus, our task was to create an ocean-bottom laser seismograph with technical characteristics, unique in terms of sensitivity and the width of the operating frequency range, and to compare the experimental data of this instrument with other laser-interference devices.

## 2. System Description

The ocean-bottom laser seismograph is a modification of a laser meter of hydrosphere pressure variations, in which frequency-stabilized semiconductor or gas helium–neon lasers are used as an emission source [14]. Figure 1 shows a scheme of the ocean-bottom laser seismograph, which operates as follows: The laser beam (17) passes through the collimator (16), in which it expands to operation-convenient size, and comes to the dividing cube (12), on which it splits into two (measuring and reference) beams. The reference laser beam travels through the system of mirrors (13 and 14) glued to the piezoceramic cylinders of excitation and compensation, and comes to the photodetector (15). The measuring laser beam then passes through the “cat’s eye” system, consisting of the lens (11) and a small reflecting mirror fixed on the membrane (10), and again enters the photodetector (15). An interference pattern forms on the photodetector, the signal from which comes to the digital registration system (18). The registration system controls the interferometer operation, using a feedback system. The main interference unit (i.e., laser, collimator, dividing cube, reflecting mirrors on piezoceramic cylinders, lens, photodetector, registration system) of the ocean-bottom laser seismograph is mounted on an optical bench (3 cm thick). The end of this optical bench is rigidly fixed to a massive lid, which closes the inner part of the ocean-bottom laser seismograph at its end. The optical bench and the lid (5 cm thick) are made of stainless steel. The lid contains a PI-100 transparent plane-parallel plate (10), a compensation chamber, and a thin membrane (9). Such arrangement of the Michelson interferometer’s optical elements minimizes the impact of external temperature variations on its operation. The compensation chamber of minimum size is located between the membrane (9) and the optical window (10). It is designed to compensate for the hydrosphere pressure variations when the instrument is being submerged or lifted. A frequency-stabilized MelesGriot helium–neon laser by with long-term stability in the ninth digit, or a single-frequency semiconductor green laser LCM-111 with a wavelength of 532 nm and long-term stability in the sixth digit, is used as a light source in the ocean-bottom laser seismograph.

The ocean-bottom laser seismograph is placed on the ocean bottom where, depending on the rocks, it is buried (sand) or rigidly fixed (granite, etc.), using various technical means. The sensing element in the ocean-bottom laser seismograph is the membrane (9). To protect the membrane from accidental damage during installation, a rubber pad (1) protects the chamber (3), where the membrane is located. The chamber contains coarse-grained sand, which provides a tight connection between the rubber pad and the membrane. The rubber pad (1) is pressed to the base of the instrument by the clamping flange (2). The clamping flange is a flat ring with specific inner and outer diameters.

When the ocean-bottom laser seismograph is being submerged, the valve (6) opens and air flows from the rubber container (7) into the compensation chamber. External hydrostatic pressure, which increases when the instrument is being submerged, presses on the container (7), and the air from this container enters the compensation chamber. As the instrument sinks, more and more air from the tank (7) enters the compensation chamber, balancing the pressure in the compensation chamber with the external hydrostatic pressure. During slow submersion, the external hydrostatic pressure is always equal to the pressure in the compensation chamber. This is done so that when the system is fully submerged on the bottom, the membrane remains in a neutral position. After full submersion with further fixing to the bottom, the valve closes and the instrument is ready for taking measurements. After completing the measuring works, the valve opens and the instrument goes up. During lifting, the air from the compensation chamber enters the container (7). This operation is performed in order to avoid damaging the membrane, and to leave it in the neutral position when lifting the instrument.

The ocean-bottom laser seismograph can operate in two modifications: In the first, power supply and information retrieval from the instrument registration system are carried out via cable lines. The instrument is controlled from the coastal observation post. In this case, it is possible to operate the ocean-bottom laser seismograph in real time, when the obtained data can be processed instantly in accordance with the assigned tasks. Another modification allows autonomous operation of the ocean-bottom laser seismograph. In this case, simultaneously with the main module, an additional module with a power supply battery is placed on the bottom. Depending on the capacity of the power supply elements, the instrument can operate for up to 10 days. Longer term operation is also possible, using more advanced power supply batteries. The information in this version is recorded on the hard media, which are located near the interferometer registration system. The obtained experimental data become accessible for processing only after lifting the instrument.

The third scheme of the ocean-bottom laser seismograph connection is associated with use of fiber-optic cable networks, which are used to solve various applied problems in large underwater areas. Moreover, it is possible to use underwater telecommunication networks, connecting not only individual territories, but also continents. Many of these cable lines, now in a semi-abandoned state, can be used for full-scale underwater monitoring.

When undertaking the first modification of the device, there are a number of both advantages and disadvantages. The advantages include a network power supply and a communication line, which enable experimental data to be obtained in real time. In addition, in this mode, it is possible to remotely control the parameters of the interferometer and remotely adjust the interference, increasing the signal-to-noise ratio which, in turn, increases the sensitivity of the ocean-bottom laser seismograph. Undoubtedly, such a scheme of the device setup plays an important role when using the ocean-bottom laser seismograph in warning systems that register catastrophic processes of various origins. The disadvantages of such an arrangement of the ocean-bottom laser seismograph include the dependence on the length of cable line and the condition of the coastline infrastructure development.

To solve problems requiring installation of the device far from the coast, the second modification is used, with a device for automating hydrophysical instruments. This device provides uninterrupted power supply to the ocean-bottom laser seismograph, and enables the recording of experimental data without interfering with the design and configuration of the seismograph.

The device consists of two parts—a registration unit and a power supply—placed in a sealed case equipped with a connection port. The registration unit includes a microcomputer, an information storage device, a multichannel data entry board, and an automatic digital converter. The power supply unit includes DC batteries, protection and coordination systems for electrical circuits, a voltage inverter, converters, and voltage stabilizers. The use of the microcomputer and multichannel data entry board enables the quick adaptation of the device to autonomous operation with various laser-interference devices.

This automation device is connected to the ocean-bottom laser seismograph by cable, through the same connector as when using a cable line from the shore. In this modification, the seismograph is set up as follows: In the location where the instrument is placed, the ocean-bottom laser seismograph is installed on the bottom first, and the automation device for hydrophysical instruments is aboard a vessel. This allows us to check the performance of the device and to correct its operation. After all of the adjustments, the device is closed and lowered onto the seabed, not far from the location of the ocean-bottom laser seismograph. When the batteries are low, the device is lifted and replaced with another one, and the laser seismograph remains on the bottom the entire time. Upon completion of the work, the automation device is lifted first, followed by the ocean-bottom laser seismograph. This is necessary in order to turn on the compensation chamber valve.

The registration system of the ocean-bottom laser seismograph measures the change in the length of the measuring arm relative to the reference arm. The change in the length of the measuring arm is due to the displacement of the membrane center, caused by the external forces associated with deformation processes at the ocean bottom at the place of the instrument’s installation. In this case, the intensity of emission coming to the photodetector is described by the following expression:(1)$$I=I_{1}+I_{2}+2\sqrt{I_{1}I_{2}}\cos\left\{\frac{4\pi \left(L_{2}-L_{1}\right)}{\lambda }\right\},$$
where *I*_1_ and *I*_2_ are intensities of the interfering beams, *L*_1_ and *L*_2_ are the optical path length of the first and second beams, respectively, and *λ* is the laser emission wavelength.

The given dependence of *I* on Δ*L* = *L*_2_ − *L*_1_ is periodic, with period *λ*/2. This dependence allows us to link to one of the extrema of the interference pattern intensity for measuring a change in the difference between the arm lengths (Δ*L*) of the interferometer, counting the interference fringes and their fraction during automatic adjustment of the interferometer to the nearest extremum of the interference pattern. The registration system provides automatic adjustment of the interferometer, and generates a signal proportional to the difference in the lengths of the interferometer arms in fractions of *λ*/2. The registration system generates a signal within Δ*l* = ±*λ*/2 under the equality Δ*l* = Δ*L* − *kλ*/2, where *k* is an integer equal to the integral part of 2Δ*L*/2. The change Δ*L* equals:(2)$$\Delta L=\frac{\lambda }{2}\left[\left(k_{+}-k_{-}\right)+\frac{U_{2}-U_{1}}{U_{\lambda /2}}\right],$$
where *k*_+_ and *k*_−_ are the number of positive and negative resets that occur when Δ*L* varies by ±*λ*/2, respectively, *U_λ_*_/2_ is the normalizing factor in volts, and *U*_1_ and *U*_2_ are the values of output voltage at the start and finish times, respectively.

For *I*_1_ = *I*_2_, the radiation intensity *I*, coming to the photodiode, is described by the following expression:(3)$$I=4I_{0}\cos^{2}\left(\frac{2\pi l}{\lambda }\right),$$
where 2*l* = 2(*L*_2_ − *L*_1_) is the optical path difference of the interfering beams.

According to (3), the value of current strength at the photodetector output is:(4)$$i=i_{0}\cos^{2}\left(\frac{2\pi l}{\lambda }\right),$$
where *i*_0_ = 4*I*_0_*χ*, and χ is the photodetector sensitivity. From (4), it follows that the change in current strength is caused by the change in the optical difference of the interferometer arms *l* and the change in wavelength *λ*. Moreover, a noise component appears, associated with the noise of the photoelectronic equipment Δ*i*_1_ and with the stability of laser radiation power Δ*i*_2_. Differentiating (4) with respect to *l* and *λ*, adding Δ*i*_1_ and Δ*i*_2_, we get:(5)$$\Delta i=i_{0}\sin\left[\frac{4\pi l}{\lambda }\right]\left\{\frac{2\pi }{\lambda }l\pm \frac{2\pi l}{\lambda^{2}}\Delta \lambda \right\}\pm \Delta i_{1}\pm \Delta i_{2},$$

Δ*i* reaches the greatest value at $\frac{4\pi l}{\lambda }=\frac{\pi }{2}$ and:(6)$$\Delta l=\frac{\Delta i}{i_{0}}\frac{\lambda }{2\pi }\pm l\frac{\Delta \lambda }{\lambda }\pm \frac{\Delta i_{1}}{i_{0}}\frac{\lambda }{2\pi }\pm \frac{\Delta i_{2}}{i_{0}}\frac{\lambda }{2\pi },$$
where $\frac{\Delta i}{i_{0}}\frac{\lambda }{2\pi }$ are seismoacoustic vibrations of the membrane, and the other components are noise.

Let us consider the noises associated with the instability of the laser emission frequency Δ*λ*/*λ*. With the instability of the laser emission frequency in the ninth digit (frequency-stabilized helium–neon laser), and when equalizing the lengths of the measuring and reference arms of the ocean-bottom laser seismograph with an accuracy of 1 cm, we find that the noises associated with frequency instability can introduce measurement error equal to *l*·Δ*λ*/*λ* = 10 pm. When using a green laser, this error increases by three orders of magnitude, and can distort measurements by the order of 10 nm.

In modern helium–neon lasers, the level of relative power fluctuations is usually a few percent, i.e., Δ*i*_2_/*i*_0_ ≅ 0.01. Consequently, the measurement sensitivity threshold is limited to a value in the order of 1 nm.

Let us estimate the sensitivity threshold determined by the shot noise of the photodetector. Ultimate sensitivity to absolute displacements in the Michelson interferometer, limited only by the photodetector shot noise, is described by the following expression:(7)$$\Delta l_{\min}=\frac{1}{4\pi }{\left\{\frac{\lambda hc\Delta f}{qP_{0}}\right\}}^{1/2},$$
where *λ* is the wavelength, *P*_0_ is the laser emission power, *c* is the speed of light, Δ*f* is the bandwidth of the received frequencies, and *q* is the photodetector’s quantum output. Assuming *P*_0_ = 0.001 W; *q* = 0.25; *c* = 3 × 10^8^ m/s; *λ* = 0.63 × 10^−6^ m; *h* = 6.626 × 10^−34^ J·s, we have:(8)$$\Delta l_{\min}=1.78\;\times \;10^{-15}\sqrt{\Delta f}\;m/{\mathrm{Hz}}^{1/2},$$

For Δ*f ≈* 10^3^–10^4^ Hz, Δ*l_min_* = 1.78 × 10^−13^ m, which does not affect the measuring accuracy.

Now, let us estimate the impacts of probable influences of external temperature on the operation of the ocean-bottom laser seismograph. When the instrument is submerged on the bottom, over time, thermal equilibrium occurs between the external temperature effect—caused by the water temperature—and the internal temperature effect, caused by the operation of the heating elements of the interferometer and the registration system. Furthermore, only an abrupt change in the external temperature can upset this thermal equilibrium. Let us roughly estimate the effect of fluctuations in the external temperature on the measuring accuracy. This effect can be realized only through the massive lid, in which the membrane is mounted, and on which the end of the optical bench is fixed. The thickness of the lid is 5 cm. For the coefficient of thermal expansion of stainless steel equal to 0.11 × 10^−4^ 1/deg, and when the temperature changes by 1 degree, the thickness of the lid will change by 0.11 × 10^−4^ × 5 × 10^−2^ = 0.55 × 10^−6^ m. Taking into account its linear expansion in all directions, we can assume that in one direction (in the direction of changing the working arm of the interferometer) this change will be about 2.8 × 10^−7^ m. In fact, these estimates are obtained from the lid, isolated from the external environment. We did not take into account the instrument housing, the air-filled inner part of the instrument, etc. We also did not account for the inertial nature of this effect and its duration, or the heat transfer coefficients of all of the components of the ocean-bottom laser seismograph.

## 3. Processing and Analysis of the Obtained Experimental Data

The ocean-bottom laser seismograph was tested in Vityaz Bay, the Sea of Japan. It was installed on the sandy bottom of the bay at a depth of about 7 m, partially buried in the soil, as shown in Figure 1. At a distance of 10 m from it, at a depth of 8 m, we placed the laser meter of hydrosphere pressure variations. The laser meter of hydrosphere pressure variations was installed on the bottom in a cubic grating, which ensured its contactless position in relation to the seabed. The distance between the bottom and the laser meter of hydrosphere pressure variations was about 0.5 m. Simultaneously, an unequal-arm laser strainmeter with a measuring arm length of 52.5 m, oriented at an angle of 18° relative to the “north–south” line—also used in hydroacoustic studies—was operated on Shultz Cape. Figure 2 shows the general layout of the instruments. In this experiment, all information from the ocean-bottom laser seismograph and the laser hydrosphere pressure variations meter came via cable lines to the onshore laboratory room, and was recorded on a computer hard disk. This computer was synchronized by a precise time system, providing measurement accuracy of 1 ms. The experimental data of the 52.5 m laser strainmeter were recorded on the hard disk of another computer, which was synchronized by the Trimble 5700 precise time system, which ensured measurement accuracy of 1 μs. Thus, we can state that the data of the ocean-bottom laser seismograph, the laser hydrosphere pressure variations meter, and the 52.5 m laser strainmeter were synchronized with accuracy of 1 ms.

In the laser meter of hydrosphere pressure variations, we installed the temperature sensors to measure temperature both inside and outside the instrument. Taking into account a small distance between the ocean-bottom laser seismograph and the laser meter of hydrosphere pressure variations, we can assume, with some approximation, that the external temperature near the ocean-bottom laser seismograph changed in the same way as near the laser meter of hydrosphere pressure variations, but with some time lag or advance. Despite this, we can assume that the external temperature of these systems changes in a similar way. Figure 3 shows the temporal change in the outside temperature, the change in hydrosphere pressure variations recorded by the laser meter, and deformation variations (vertical displacements) of the upper layer of the oceanic crust, recorded by the ocean-bottom laser seismograph.

When comparing the data shown in Figure 3, the general behavior of hydrosphere pressure variations (Figure 3b) and vertical displacements of the oceanic crust is similar. There is some similar general behavior of hydrosphere pressure variations and external temperature. As we can see in Figure 3, the external temperature variations affect the readings of the laser meter of hydrosphere pressure variations. At the same time, these variations practically do not affect the readings of the ocean-bottom laser seismograph. This is due to the influence of the external temperature on the membrane deformation in the case of the laser meter of hydrosphere pressure variations, and the weak influence of external temperature variations on the deformation of the membrane buried in the sand of the oceanic crust’s upper layer.

An interesting issue is the registration of vertical displacements of the Earth’s crust, caused by the hydrosphere pressure variations during the passage of progressive sea gravity waves (wind waves). In the period of testing the ocean-bottom laser seismograph, there were no powerful sources of wind waves. During the observation period, we recorded background wind (gravity) waves with periods from 5 to 7 s (with minor variations). The laser meter of hydrosphere pressure variations in water and the 52.5 m “north–south” laser strainmeter at Shultz Cape were operating simultaneously. The waves arriving at Schultz Cape, according to [15], at depths equal to a half-length of a gravity wave (in our case, with a period of 6.2 s, this was approximately equal to 30 m), began to interact intensively with the bottom, transferring a part of the energy into it, which propagated further into the Earth’s crust in the form of primary microseisms. The 52.5 m laser strainmeter recorded these microseisms. A propagating sea wind wave (swell) entered Vityaz Bay; its front turned due to refraction, and moved in the direction of the ocean-bottom laser seismograph and the laser meter of hydrosphere pressure variations, which registered the vertical displacements of the seabed and hydrosphere pressure variations, respectively, caused by the passing wind wave. Figure 4 shows the spectra obtained during the processing of synchronous record fragments of the laser strainmeter in the microseismic range (a), the ocean-bottom laser seismograph (b), and the laser meter of hydrosphere pressure variations (c). The duration of the processed fragment was about 8.5 min. Taking into account the propagation speed of the sea wind wave with a period of 6.2 s, which was in the range of 5–10 m/s, along with distance covered by it from the bay entrance to the location of the instruments, we can assume that we were processing almost-synchronous record fragments. In the spectrum of the ocean-bottom laser seismograph record, a peak with period of about 5.9 s stands out; the same peak can be identified in the record of the laser meter of hydrosphere pressure variations. As the wind wave moved, its period decreased from 6.2 to 5.9 s due to conversion of a part of its energy into the energy of the upper layer of the Earth’s crust, the physics of which are described in [15].

With such an arrangement of the ocean-bottom laser seismograph and the laser meter of hydrosphere pressure variations, we cannot confirm that the ocean-bottom laser seismograph registers the vertical component of the Rayleigh wave, which formed as a result of transformation of the sea wind wave into the primary Rayleigh-type microseisms. We can only assert that the pressure variations of 4.5 Pa caused vertical displacements of 0.11 μm in the upper layer of the Earth’s crust at the location of the ocean-bottom laser seismograph.

At the 52.5 m laser strainmeter installed at Shultz Cape, the displacement of the strainmeter base was 2.2 nm at the frequency corresponding to the wind wave period of 6.2 s. When using a classical laser strainmeter, such as the one used in this experiment, the recorded amplitude relates to the true amplitude of the wave propagating along the upper layer of the Earth’s crust, given by the following equation:(9)$$A=2A_{0}\sin^{2}\left(\frac{kL}{2}\right)=2A_{0}\sin^{2}\left(\frac{\pi L}{\lambda }\right),$$
where *A*_0_ is the wave amplitude projection onto the strainmeter axis, *k* = 2*π*/*λ* is the wave number, *λ* is the wavelength, *ω* = 2*πν* is the circular frequency, *ν* is the wave frequency, *t* is the current time, and *L* is the length of the strainmeter’s working arm. If we make calculations for deep water, then *λ* = 60 m. Then, from (9), we can find that *A*_0_ = 7.5 nm. This is the horizontal component of the Rayleigh wave, which propagates along the axis of the 52.5 m laser strainmeter mounted perpendicular to the coastline. This statement can be confirmed by the fact that another laser strainmeter, with an arm length of 17.5 m, the axis of which is located along the coastline, did not register a disturbance with a period of 6.2 s. If we assume that the ocean-bottom laser seismograph registers the vertical component, and the 52.5 m strainmeter registers only a part of the horizontal component, then the Rayleigh wave ellipse should be located at an angle of 86° to the horizon.

Just 60 m from the laser strainmeter with a measuring arm length of 52.5 m, at a depth of 3 m a Guralp CMG-3ESPB broadband velocimeter was installed, consisting of three sensors that measure soil vibrations simultaneously in three (north–south, east–west, and vertical) directions. The working range of each sensor is 0.003–50 Hz. As we said earlier, the ocean-bottom laser seismograph mainly measures the vertical displacement of the bottom particles; therefore, it is better to use the data of the vertical component of the velocimeter to compare data from the ocean-bottom laser seismograph and from the velocimeter. Let us compare the experimental data from the ocean-bottom laser seismograph with the data from the vertical sensor of the velocimeter. If we consider the synchronous fragments of the instruments’ records from 23 July 2021, starting at 6:32:17, the duration of each fragment is two hours. First, let us filter the experimental data from the ocean-bottom laser seismograph with Hamming bandpass filter in the range from 3 mHz to 50 Hz, and then sample the data down 10 times, with averaging. As a result, we get the graph shown in Figure 5a,b, showing the record of the vertical sensor of the velocimeter.

Let us analyze the spectra of the ocean-bottom laser seismograph and the vertical sensor of the velocimeter records. The spectra show peaks with the same periods in the range of the wind waves’ periods. Thus, the spectrum of the ocean-bottom laser seismograph (Figure 6a) contains peaks with periods of 5.6 s, 5.1 s, and 4.5 s. On the spectrum of the velocimeter record (Figure 6b) there are peaks with periods of 5.1 s and 4.5 s. Moreover, on the spectrum of the laser seismograph, the amplitude of oscillations with a period of 5.1 s is greater than the amplitude of neighboring oscillations, and on the spectrum of the velocimeter record, the oscillations with period of 4.5 s have the largest amplitude. The difference in the amplitudes of these oscillations is associated with the location of the instruments: the seismograph was installed on the bottom of the bay, while the velocimeter was located at a distance of about 700 m from the seismograph, and 350 m from the coastline of Vityaz Bay. When analyzing the recording spectrum of the laser meter of hydrosphere pressure variations, we identified peaks with periods of 5.1 and 5.5 s, corresponding to wind waves. From this connection, we can conclude that the source of the 5.1 and 5.6 s oscillations, recorded by the ocean-bottom laser seismograph was the wind waves of Vityaz Bay.

Next, let us dwell on the registration of lower frequency oscillations that exist in Vityaz Bay. These oscillations, first of all, include the eigenoscillations of the bay (seiches), which we recorded earlier, with a main period of about 16–18 min [16]. Seiches, interacting with the bottom, excited the elastic vibrations of the corresponding period, which were recorded by the 52.5 m laser strainmeter installed at Shultz Cape [16]. To study the capabilities of the ocean-bottom laser seismograph in registering such disturbances, we processed the synchronous record fragments of the ocean-bottom laser seismograph, the laser strainmeter, and the laser meter of hydrosphere pressure variations. Figure 7 shows these record fragments, while Figure 8 shows spectra obtained from processing the records of the ocean-bottom laser seismograph, the 52.5 m laser strainmeter, and the laser meter of hydrosphere pressure variations, as shown in Figure 7.

As we can see in Figure 7, the records of the ocean-bottom laser seismograph and the 52.5 m laser strainmeter evidently contain low-frequency oscillations of approximately the same period. At the same time, in the record of the laser meter of hydrosphere pressure variations, the oscillations of approximately the same periods are evident in the second half of the record. In the first half, the record is exposed to the effects of other low-frequency processes associated with possible influence on the membrane of the trains of short-period internal waves, which were confidently recorded by the temperature sensor installed on the laser meter of hydrosphere pressure variations. A similar reaction to these disturbances can be identified in the two upper diagrams in Figure 4. Furthermore, the record fragments shown in Figure 7 were subjected to spectral processing. Some fragments of the obtained spectra are shown in Figure 8. In the first two spectra (Figure 8a,b), the second maximum has a period of 17 min and 4 s. In Figure 8a, the height of this maximum is 0.34 µm (for the ocean-bottom laser seismograph), and for the 52.5 m laser strainmeter it is 0.6 µm (see Figure 8b). This peak is not explicitly evident in the spectral record of the laser meter of hydrosphere pressure variations, but we can observe a broadening of the peak corresponding to the period of 27 min and 18.4 s, with an amplitude of 50.8 Pa. This peak “masked” the peak with the period of 17 min and 4 s. Because of poor frequency resolution, due to the length of the processed fragment, the more power-intensive peak “absorbed” the less power-intensive peak. The spectra were obtained using the periodogram method, which is based on the fast Fourier transform. If we process only the right side of the record of the laser meter of hydrosphere pressure variations via the periodogram method, then the peak with the period of 17 min and 4 s, with and amplitude of 18 Pa, will stand out in the spectrum; this spectral fragment is shown in Figure 9.

In the conclusion of this section, we can note that any change in hydrospheric pressure (e.g., tides, surge phenomena, sea gravity and infragravity waves, hydroacoustic oscillations, and waves of artificial and natural origin) causes displacements (deformations) in the seabed at the corresponding oscillation periods. The contribution of each signal (wave) can be estimated only during subsequent spectral processing, by belonging to a particular frequency range. Given that the ocean-bottom laser seismograph (the sensitive element of which is a membrane) was located in the seabed, we can assume that it registered the displacement of ocean-bottom particles, caused by the impact of variable hydrospheric pressure created by various wave and non-wave processes of artificial and natural origin.

## 4. Conclusions

As a result of the discovered technical solutions, the laser meter of hydrosphere pressure variations was transformed into the ocean-bottom laser seismograph, designed to measure the vertical seabed displacements at the place of its location. The accuracy of measuring vertical displacements is limited by the stability of laser emission power, and is equal to about 1 nm, which can be improved by using lasers with better technical characteristics. The ocean-bottom laser seismograph is designed to measure vertical displacements of the bottom in a wide frequency range. The upper frequency is determined by the operating speed of the digital registration system, and is equal to 1000 Hz, while the lower frequency is determined by the membrane’s elastic characteristics and its elastic bond with the seabed particles at the location of the instrument. Theoretically, the lower frequency is determined by the duration of the measurements, and is equal to 1/t, where t is the duration of the measurements in seconds. The efficiency of using the ocean-bottom laser seismograph was confirmed by registering elastic seabed vibrations caused by sea gravity waves with periods of about 6 s, as well as the waves caused by eigenvibrations of the bay at the instrument location, the main period of which was 17 min and 4 s. Comparison of the experimental data spectra of the ocean-bottom laser seismograph with the experimental data of the laser hydrosphere pressure variations meter and the velocimeter in the range of wind wave periods revealed harmonics with the same periods. Elastic oscillations of the seabed with periods of 5.1 s and 5.6 s were caused by wind waves in the bay where the instruments were installed. The compensation system of the ocean-bottom laser seismograph allows it to be installed on the bottom at depths of up to 50 m. Considering that the Arctic seas have shallow average depths, this ocean-bottom laser seismograph can be used there with great success. The ways to improve the technical characteristics of the ocean-bottom laser seismograph are associated with improvement of the pressure compensation system during submersion and lifting, and with improvement of the methods of its installation at the bottom, ensuring strong contact between the membrane and the seabed particles. Using a forced compensation system for hydrospheric pressure in the compensation chamber, the ocean-bottom laser seismograph can be installed on the bottom at depths of up to 4000 m. In addition, its installation is possible with use of additional technical means and unmanned underwater vehicles.

Further use of the ocean-bottom laser seismograph in conjunction with a laser nanobarograph, a laser strainmeter, and a laser meter of hydrosphere pressure variations makes it possible to determine the primary source of certain vibrations. The use of laser-interference devices in the geospheric transition zone makes it possible to determine the primary source of vibrations manifested in two or more neighboring geospheres. The primary source can be determined by the time of arrival of the signal, the amplitude of the oscillations, the ratio of the studied oscillations to background noise, etc. Simultaneous use of an ocean-bottom laser seismograph will allow us to assess the contribution of sea waves to the vertical component of elastic vibrations of the upper layer of the Earth’s crust.

## Figures and Tables

**Figure 1 sensors-22-02527-f001:**
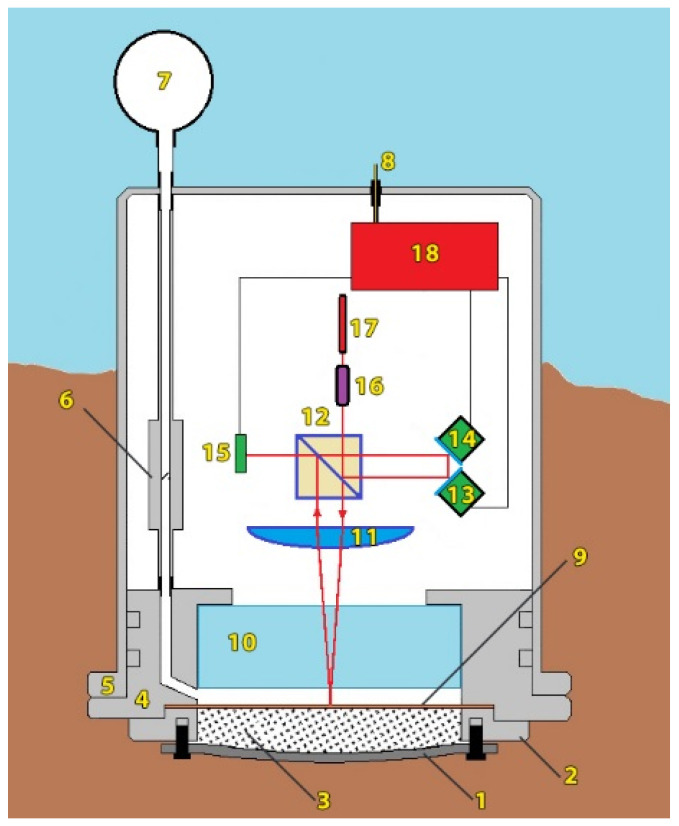
Bottom-type laser-interference system: 1—rubber pad, 2—clamping flange, 3—additional module, 4—instrument lid, on which the interferometer is mounted, 5—instrument case, 6—valve, 7—air-filled container, 8—pressure seal for signal and power cables, 9—membrane, 10—optical window (PI-80), 11—lens, 12—dividing cube, 13—piezoceramic transducer of excitation, 14—piezoceramic transducer of compensation, 15—photodetector, 16—collimator, 17—laser, 18—digital registration system.

**Figure 2 sensors-22-02527-f002:**
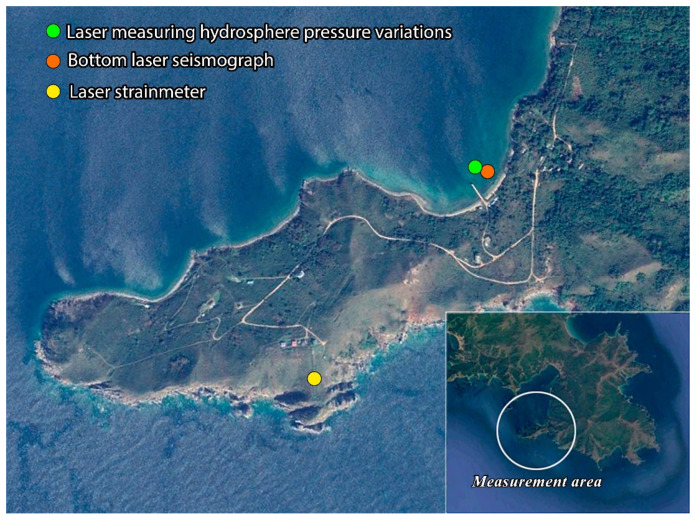
General layout of the instruments.

**Figure 3 sensors-22-02527-f003:**
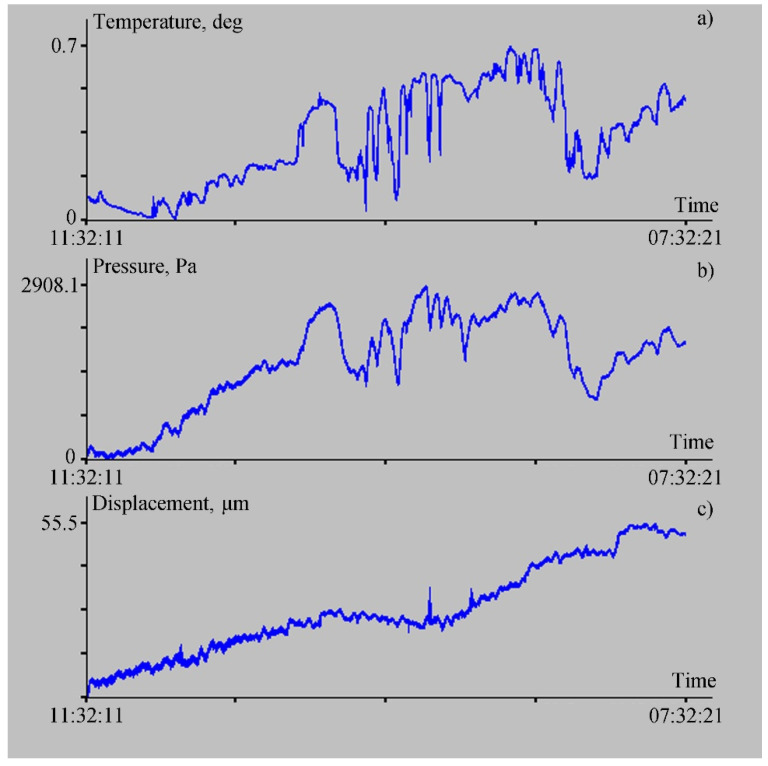
Temporal changes in external temperature variations (**a**), hydrosphere pressure variations (**b**), and vertical displacements of the Earth’s crust segment (**c**) for 21 July 2017.

**Figure 4 sensors-22-02527-f004:**
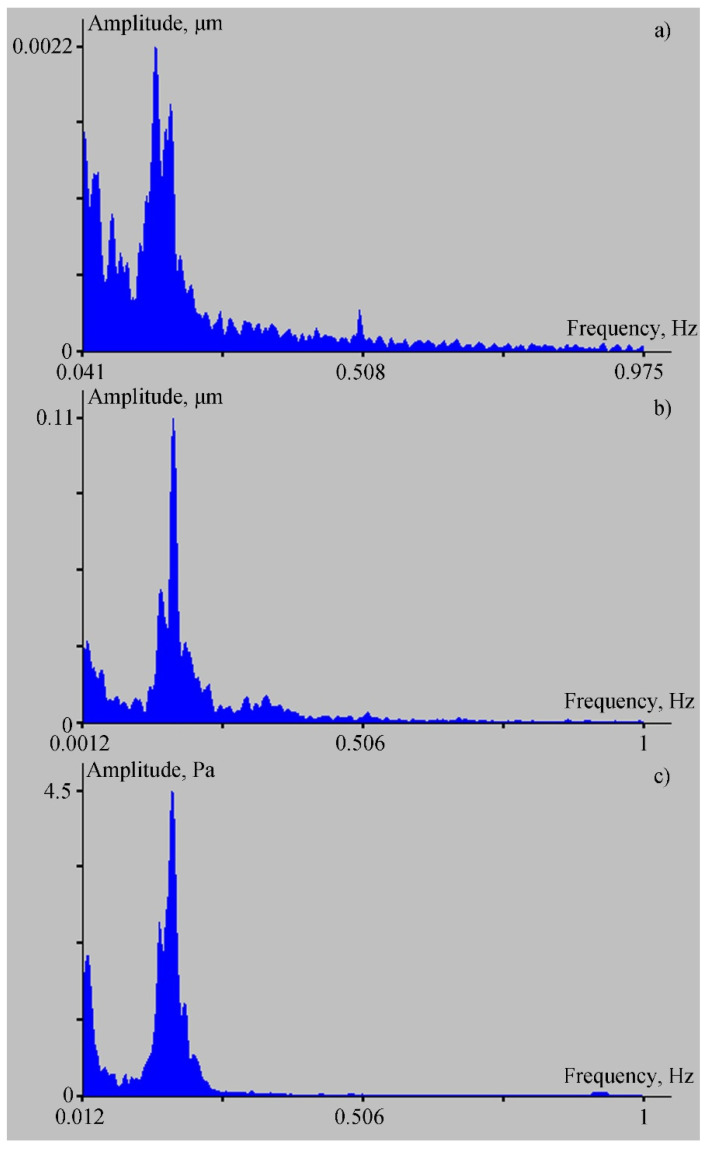
Spectra obtained during processing of synchronous record fragments in the microseismic range of the laser strainmeter (**a**), the ocean-bottom laser seismograph (**b**), and the laser meter of hydrosphere pressure variations (**c**).

**Figure 5 sensors-22-02527-f005:**
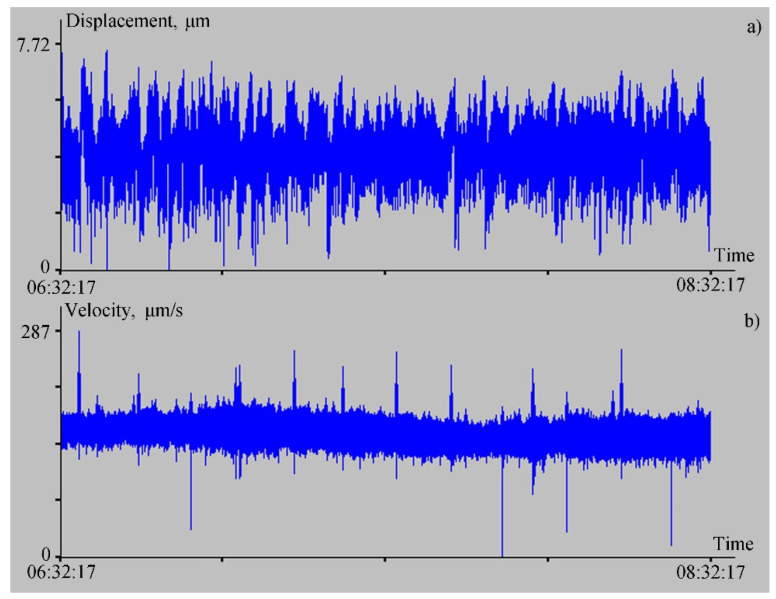
Filtered recording of the ocean-bottom laser seismograph (**a**) and the record of the vertical sensor of the velocimeter (**b**) for 23 July 2021.

**Figure 6 sensors-22-02527-f006:**
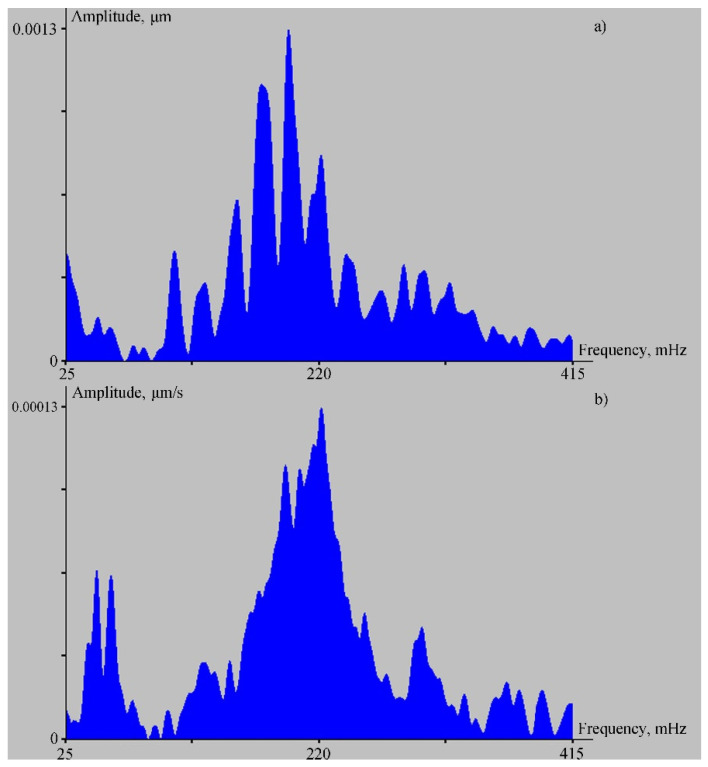
Spectra obtained from processing the records of the ocean-bottom laser seismograph (**a**) and the vertical sensor of the velocimeter (**b**) shown in Figure 5.

**Figure 7 sensors-22-02527-f007:**
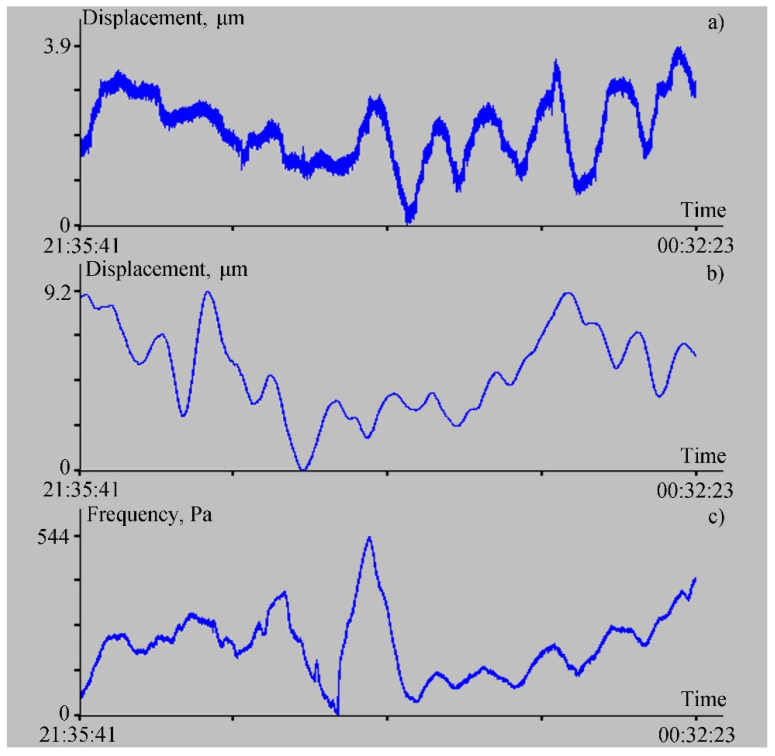
Synchronous record fragments of the ocean-bottom laser seismograph (**a**), the 52.5 m laser strainmeter (**b**), and the laser meter of hydrosphere pressure variations (**c**) for 23–24 July 2021.

**Figure 8 sensors-22-02527-f008:**
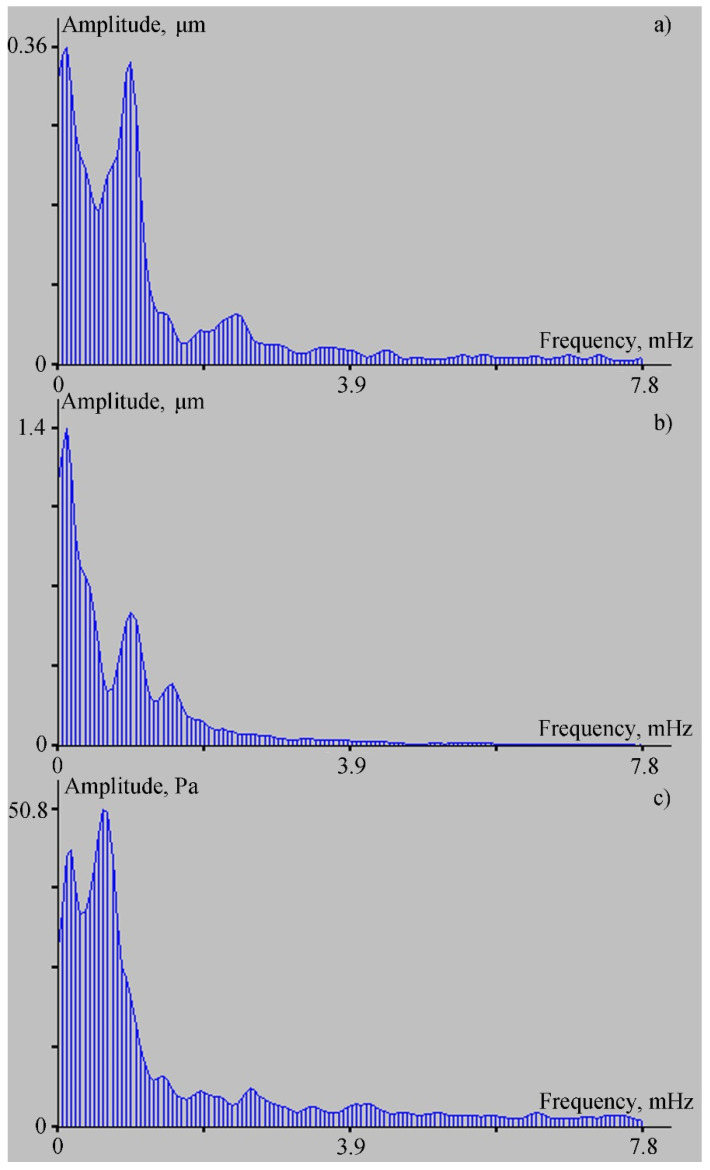
Spectra obtained from processing the records of the ocean-bottom laser seismograph (**a**), the 52.5 m laser strainmeter (**b**), and the laser meter of hydrosphere pressure variations (**c**), as shown in Figure 7.

**Figure 9 sensors-22-02527-f009:**
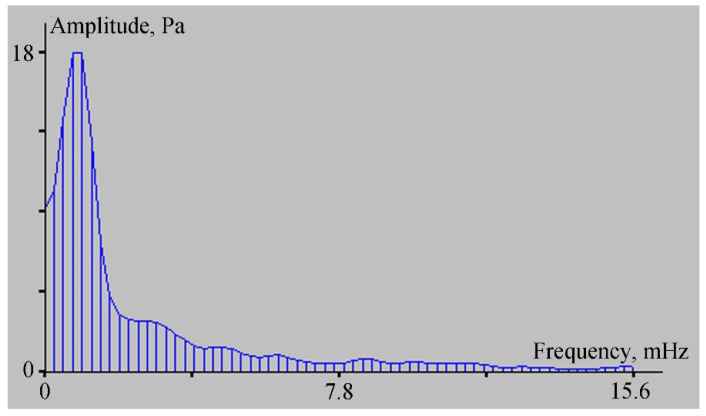
Spectrum of the record fragment of the laser meter of hydrosphere pressure variations.

## Data Availability

Third-party data; restrictions apply to the availability of these data.

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
