# Peer review of "Ocean-Bottom Laser Seismograph"

_sensors, 2022, doi:10.3390/s22072527_

Round 1

Reviewer 1 Report

1. The description of device1,2,3 and 7 in Figure 1, can you   precisely delineate  their  relationship  and the Workflow?

2. When the equipment is placed offshore, the change of sea water static pressure caused by tide relative to water depth can not be ignored. The change of static pressure of the waves and the acoustic pressure will change the displacement of device1 at the same time .  How to distinguish the difference between sea water static pressure change by the waves and seabed vertical displacement by the acoustic pressure?

3. The principle of the  equipment  has been introduced clearly, but how to modify the whole system and the two optimization points? What are the benefits of these optimization ?

4. Can the three instruments in Figure 2 observe synchronously?  and What measures for the time accuracy should be taken ?

5.  Horizontal and vertical axis unit in Figure 6 has not been defined clearly.

Author Response

Thank you for reviewing the article. The answers to the comments are in the attached file.

Reviewer 2 Report

Authors proposed ocean-bottom laser seismograph, based on the modified laser meter of hydrosphere pressure variations. The operating frequency range is 1 kHz. However, authors need to use professional English services or native English colleague professors because there are many broken English grammar mistakes. In addition, authors need to provide in detail analysis for the results. Therefore, the manuscript need to be revised accordingly.

  1. Authors need to provide some important data in the abstract and conclusion sections.
  2. In Line 276, figure 3b -> Figure 3b. Please check others.
  3. In Line 278, figure -> Figure. Please check others.
  4. In Figure 3c, what is y-axis information ?
  5. In Figure 3, c information is missing.
  6. In Figure 4, authors need to provide information in x- and y-axes.
  7. Figure 6 labels are too small to be seen.
  8. Please use formal English expression. And -> In addition.
  9. Authors need to use abbreviated journal names in the reference sections.
  10. In the introduction, authors must emphasize the novelty of the manuscript.

Author Response

(The authors gave the same response as above.)
